# On the Malleability of Consumer Attitudes toward Disruptive Technologies: A Pilot Study of Cryptocurrencies

**Horst Treiblmaier** [1,*] and **Evgeny Gorbunov** [2]

1   School of International Management, Modul University Vienna, 1190 Vienna, Austria
2   Independent Researcher, Rossauer Lande 47-49, 1090 Vienna, Austria; eg@disence.com
*   Correspondence: horst.treiblmaier@modul.ac.at

**Abstract:** The digital transformation of core marketing activities substantially impacts relations between consumers and companies. Novel technologies are usually complex, making their underlying functionality as well as the desirable and undesirable implications hard to grasp for ordinary consumers. Cryptocurrencies are a prominent yet controversial and poorly understood example of an innovation that may transform companies' future marketing activities. In this study, we investigate how easily consumers' attitudes toward cryptocurrencies can be shaped by splitting a convenience sample of 100 consumers into two equal groups and exposing them to true, but biased, information about cryptocurrencies (including market forecasts), respectively, highlighting either the advantages or disadvantages of the technology. We subsequently found a significant difference in the trust, security and risk perceptions between the two groups; specifically, more positive attitudes pertaining to trust, security, risk and financial gains prevailed in the group exposed to positively-skewed information, while perceptions regarding trust, risk and the sustainability of cryptocurrencies were weaker among the group exposed to negatively-skewed information. These findings reveal some important insights into how easily consumer attitudes toward new technologies can be shaped through the presentation of lopsided information and call for further in-depth research in this important yet under-researched field.

**Keywords:** digital transformation; blockchain; cryptocurrencies; consumer attitudes; trust; security; privacy; perceived risk; financial gains; sustainability; experiment

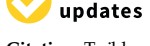



## 1. Introduction

Applications based on blockchain technologies, including cryptocurrencies, are predicted to fundamentally change current marketing strategies and activities [1,2]. They consist of multiple components [3], and their underlying technological foundations are complex and hard to understand for a non-technical audience [4]. Consequently, individuals who buy, trade and use cryptocurrencies, frequently need to rely on superficial knowledge as the basis of their decision-making.

Given the importance of the global cryptocurrency market, whose total revenue was 1543 million USD in 2021 and is expected to reach 2303 million USD by 2028 [5], the question arises what the underlying drivers of individuals' decision making are and how easily they can be changed. This is especially important for companies that plan to deploy cryptocurrencies and need to know how they are perceived by consumers [6]. According to Yoon et al. [7], "individuals tend to selectively rely on information that corresponds with their personal attitudes and decisions" (p. 93). In general, it is crucial for companies to carefully deploy technological innovations, which may have detrimental side effects for their target audience.

For example, the storage and use of personal data can enhance the quality of individualized offerings, however, can also result in consumers being relentlessly targeted without their explicit approval or even becoming victims of data breaches without any chance to

intervene [8]. From a commercial perspective, marketers thus need to diligently communicate about the use and potential advantages of technological innovations to those who are affected. Cryptocurrencies possess numerous features that are perceived as either positive or negative, and they affect individuals, organizations and nations in various respects. Examples for such controversial topics include traceability, privacy and security [9].

Substantial academic research exists that investigates how individuals build and change their attitudes over time. Previous research found that consumer attitudes toward technologies are shaped, among other factors, by their (a) knowledge of the technology, (b) previous experiences and (c) perceptions of the expected benefits [10]. Thus far, however, no study exists that specifically investigates how easily individuals' attitudes toward cryptocurrencies can be influenced. In this pilot study, we therefore strive to answer the following research question: *How easily can relevant consumer attitudes toward cryptocurrencies be shaped through communications?*

## 2. Theoretical Background

Technological change rarely happens without controversy and the revelation of various positive and negative consequences. Blockchain technology provides an example that is predicted to transform future marketing activities by fostering disintermediation, combating click fraud, reinforcing trust and transparency, enhancing privacy protection, empowering digital marketing security and enabling creative loyalty programs [2]. The academic marketing literature has already acknowledged the potential of blockchain to solve important pending issues, for example by enabling increased control over transactional data, enhanced supply chain coordination and improved data privacy [11–13].

Blockchain is a layered combination of various independent technologies that are currently still under development. It still faces numerous challenges, such as lack of interoperability [14] and the exploitation of the full potential of smart contracts [15]. Blockchain technology was popularized by the release of the source code of Bitcoin in 2009, representing the first digital asset enabling the online transfer of value without intermediaries that can be designated as programmable money.

The media coverage of Bitcoin and cryptocurrencies in general, remains highly controversial and often tends to be skewed in either a positive or negative direction. Given the complexity of the underlying technology, few end users are fully aware of the inner workings of the technological underpinnings and thus recognize the resulting characteristics, such as immutability and transparency of data [3]. They therefore have to rely on companies and their brands to transparently signal the responsible use of the technology.

From a technical perspective, cryptocurrencies (i.e., digital currencies that run on a decentralized system) can have their own native blockchain (e.g., Bitcoin), in which case they are called coins. Conversely, so-called tokens, which are created on an already existing blockchain, offer a multitude of different services and are predicted to rapidly gain in popularity [16]. Since the overarching term cryptocurrency is more widely known and also encompasses a wide range of functionality, including payments, financial investments and various types of utilities, we use this term in our study.

Existing applications already hint at the potential benefits of cryptocurrencies and the resulting implications for marketing. For example, consumers might financially benefit from reduced transaction costs and advanced personalization services of blockchain-based loyalty programs; however, the processing and storing of personal information might simultaneously open up new privacy risks that need to be taken care of [17].

Previous research has identified several important constructs that shape consumer attitudes toward cryptocurrencies, namely security, trust, privacy, perceived risk, financial gains and sustainability. Encryption is used to verify the integrity and authenticity of transactions, which offers new security standards.

While frequently labeled as "trustless technologies", consumers still need to trust the underlying technology, the cryptography, the source code and also the validators. Cryptocurrencies offer different levels of (perceived) privacy, with many of them being

pseudonymous, which means that addresses (e.g., a hashed version of a public key) are stored on public ledgers. Similarly, the sustainability of cryptocurrencies is controversial, with the energy consumption needed for the Proof-of-Work consensus mechanism (as is used in Bitcoin's Nakamoto consensus) being a major point of criticism.

Conversely, cryptocurrency proponents have indicated environmental benefits that might arise from a more widespread adoption. Cryptocurrencies are often seen as an investment instrument, and financial gains are a major motivator for buying and trading them. Taken together, their perceived attributes create a level of (perceived) risk that ultimately depends on an individual's personal assessment [1,2,18–20].

From a theoretical perspective, the topic of attitudinal malleability (i.e., how easily attitudes can be altered by external forces) has been mainly researched in the context of an overarching epistemological understanding of the world. In general, there is some disagreement on whether an individual's character, competence and attitude is rather fixed (entity theory) or malleable (incremental theory) [21]. The importance of a theorist's respective viewpoint on the assessment of actual behavior was confirmed in previous research [22].

In a marketing context, the self was shown to be partly stable and partly malleable, and brands can benefit from a thorough understanding of how consumer attitudes can be influenced [23]. In a study investigating the malleability of implicit gender attitudes, the authors found that their experimental intervention (i.e., a randomized video priming treatment) had only limited effects among a specific subpopulation in Tunisia [24]. In another research project examining the malleability of ageist attitudes, the authors found that exposure to pictures of either admired or disliked young and old individuals impacted the study participants' attitudes [25].

In a more recent study investigating pre-service teachers' attitudes toward statistics, the authors found a change in attitudes after the teachers completed an educational course in reform statistics [26]. To sum up, previous results are conflicting and indicate that the malleability of attitudes depends on the subject matter, the trigger for attitudinal change and the respective population. The investigation of attitudinal malleability toward innovative technologies, and especially cryptocurrencies, therefore, constitutes a pending research gap. Specifically, the question arises of whether the relevant attitudinal constructs identified in the literature are subject to attitudinal malleability [27].

## 3. Methodology

The goal of our study is to discover whether individuals' attitudes toward cryptocurrencies can be changed within a relatively short period of time through exposure to positively- or negatively-skewed information. Specifically, we tested for effects on consumer perceptions regarding trust, security, privacy, perceived risk, financial gains and sustainability, which were identified as highly important attitudinal and behavioral antecedents in previous research.

To be able to answer the research question, we used an experimental design and measured the attitudes of the study participants before and after exposure to a short video that outlined various positive (group "advantages") and negative (group "disadvantages") aspects of cryptocurrencies, respectively. The videos also included forecasts published by market researchers [28] and did not necessarily present the information in a balanced manner (i.e., in a way such that the absolute positive or negative impact was assessed in an objective way). In our study, we purposefully focused on cryptocurrencies rather than tokens, since the former term is more popular.

The scales used to measure attitudes were identical for the pre- and post-treatment measurements and can be found in Appendix A, and the links to the respective videos can be found in Appendix B. All items are based on previous literature, yet were modified to fit the purpose of our research [29] and to ensure content validity and understandability. The participants were randomly assigned to one of two groups, each of which watched a short video (~2.5 min) that had been produced by the researchers exclusively for this study.

Each of the two videos contained factually correct, yet unbalanced information in order to highlight either the advantages or disadvantages of cryptocurrencies pertaining to the six constructs under investigation. Our convenience sample was recruited online using a snowballing system and randomization was used to control for unknown confounding variables. The experiment was conducted in April and May 2021.

We collected a total of 100 responses, having assigned 50 participants to each group. The constructs were measured using 10-point Likert-type scales with end-points of 1 ("totally disagree") and 10 ("totally agree"). The data analysis was performed with RStudio 1.4.1717. Construct scores were calculated for each participant by adding up the scores of the respective items and calculating the average. We use unpaired *t*-tests to investigate whether any significant differences could be detected between the two groups and paired *t*-tests to uncover attitudinal changes within the groups as a result of exposure to the stimuli.

## 4. Results

The demographics of the total sample are shown in Table 1. Using a between-subjects design, 50 participants were exposed to information pertaining to potential advantages of cryptocurrencies and the other 50 to potential disadvantages. Our sample was biased toward a male population as well as younger and better-educated respondents. Two thirds of the respondents actually owned cryptocurrencies, which is above the population average and suggests that our target group may have an above-average understanding of the subject matter. We have little evidence for a response bias given that numerous differences in our study were not statistically significant between the "pre" and "post" measurements.

**Table 1.** Demographic data (*n* = 100).

| Gender | Male | Female | |
|---|---|---|---|
| | 65% | 35% | |
| Age | 18–24 | 25–35 | 35+ |
| | 46% | 26% | 28% |
| Cryptocurrency ownership | Yes | No | |
| | 67% | 33% | |
| Occupational status | Student | Employed | Self-employed |
| | 52% | 42% | 6% |

For all constructs, the information presented in the videos had the predicted impacts, such that positive information improved individuals' attitudes and vice versa (see Figure 1); however, not all of the differences between the groups, or between the measurements within each group, were statistically significant at the 0.05 level (see Tables 2 and 3). The unpaired *t*-tests revealed no significant differences in the construct means between the two groups prior to their exposure to the respective videos, which indicates that the randomized sampling process was successful (see the "Pre (1)" section in Table 2).

In the following sections, we discuss the respective constructs in more detail. The subscripts "1" and "2" denote the attitude evaluations that were administered before (pre) and after (post) exposure to the stimuli. "a" and "b" denote advantages and disadvantages, respectively. We start the discussion of each construct with a brief intergroup comparison (i.e., advantages vs. disadvantages, see "post (2)" in Table 2) followed by an intragroup comparison (i.e., pre vs. post, see Table 3). The standard deviations are given in brackets, and the confidence intervals are shown in Figure 1.

### 4.1. Trust

Previous research has established trust in innovative technologies as a complex construct that may have huge implications for consumer behavior [30]. Our measurement items for trust incorporate a general assessment of cryptocurrencies as well as the perception of currency exchanges, which serve as crucial on- and off-ramps that are needed to

convert fiat money into cryptocurrencies and vice versa. Furthermore, we consider the fact that cryptocurrencies are not yet fully regulated.

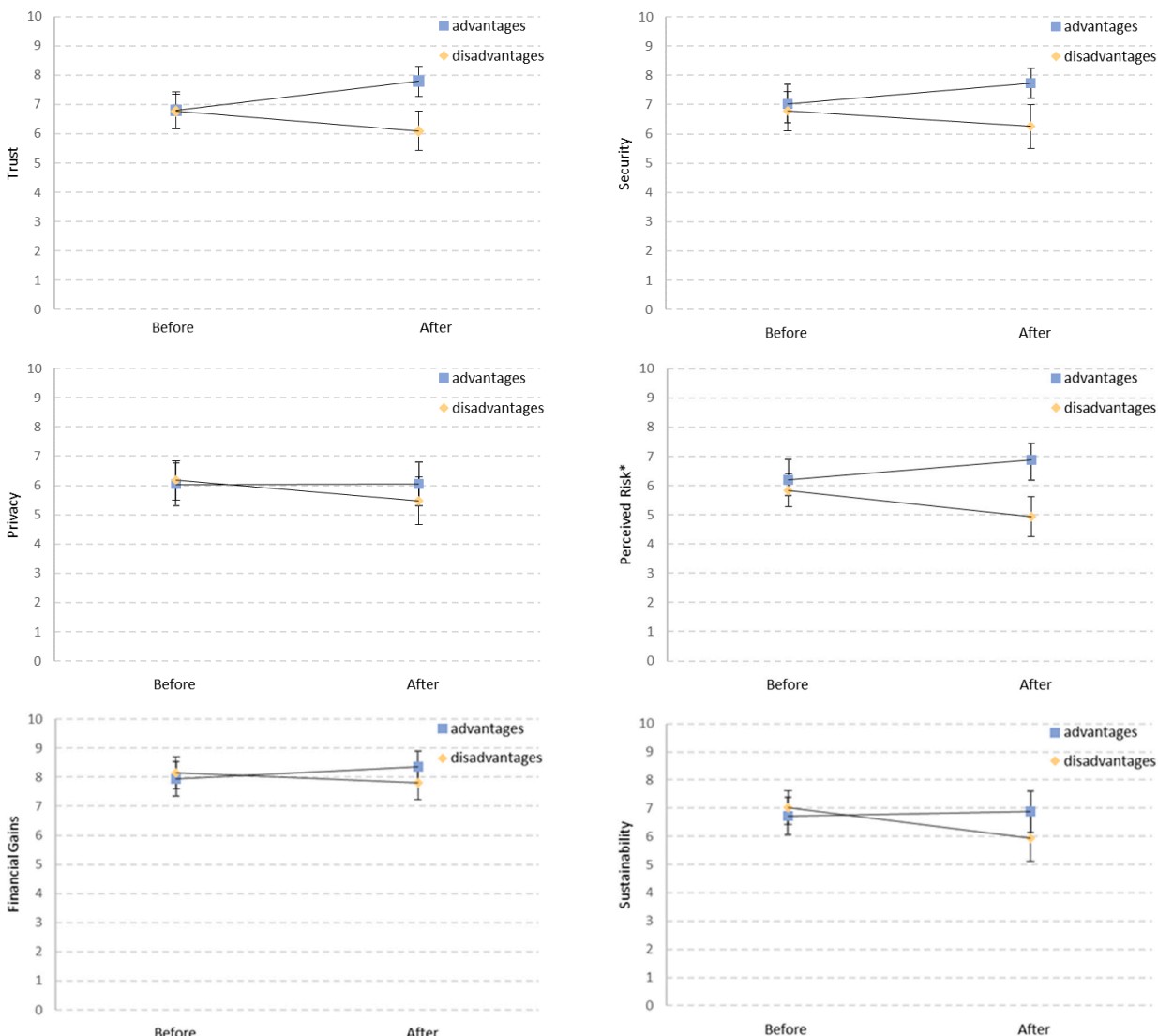

**Figure 1.** Changes in attitudes after exposure to the informational videos (vertical lines representing confidence intervals) * reverse coded.

**Table 2.** Intergroup comparisons before and after the stimulus.

| | Pre (1) | | | Post (2) | | |
|---|---|---|---|---|---|---|
| | **Advantages** | **Disadvantages** | **Unpaired *t*-Test** | **Advantages** | **Disadvantages** | **Unpaired *t*-Test** |
| Trust | 6.81 (2.24) | 6.77 (2.14) | $t = 0.11$ ($p = 0.92$) | 7.79 (1.84) | 6.10 (2.35) | $t = 4.01$ ($p < 0.05$) |
| Security | 7.03 (2.31) | 6.78 (2.33) | $t = 0.53$ ($p = 0.59$) | 7.74 (1.83) | 6.26 (2.65) | $t = 3.25$ ($p < 0.05$) |
| Privacy | 6.05 (2.59) | 6.18 (2.35) | $t = -0.26$ ($p = 0.79$) | 6.06 (2.65) | 5.47 (2.87) | $t = 1.07$ ($p = 0.29$) |
| Perceived Risk | 6.21 (2.38) | 5.84 (1.97) | $t = 0.85$ ($p = 0.40$) | 6.87 (2.04) | 4.93 (2.42) | $t = 4.34$ ($p < 0.05$) |
| Financial Gains | 7.95 (2.07) | 8.15 (1.91) | $t = -0.50$ ($p = 0.61$) | 8.37 (1.88) | 7.80 (1.98) | $t = 1.47$ ($p = 0.14$) |
| Sustainability | 6.72 (2.33) | 7.02(2.12) | $t = -0.67$ ($p = 0.50$) | 6.88 (2.57) | 5.93 (2.83) | $t = 1.75$ ($p = 0.08$) |

Mean values and standard deviation in brackets, and significant results are highlighted.

**Table 3.** Intragroup comparison before and after the stimulus.

| | Advantages | | | Disadvantages | | |
|---|---|---|---|---|---|---|
| | **Pre (1)** | **Post (2)** | **Paired *t*-Test** | **Pre (1)** | **Post (2)** | **Paired *t*-Test** |
| Trust | 6.81 (2.24) | 7.79 (1.84) | $t = -4.62$ ($p < 0.05$) | 6.77 (2.14) | 6.10 (2.35) | $t = 2.55$ ($p < 0.05$) |
| Security | 7.03 (2.31) | 7.74 (1.83) | $t = -3.13$ ($p < 0.05$) | 6.78 (2.33) | 6.26 (2.65) | $t = 1.73$ ($p = 0.09$) |
| Privacy | 6.05 (2.59) | 6.06 (2.65) | $t = -0.02$ ($p = 0.98$) | 6.18 (2.35) | 5.47 (2.87) | $t = 1.62$ ($p = 0.11$) |
| Perceived Risk | 6.21 (2.38) | 6.87 (2.04) | $t = -2.58$ ($p < 0.05$) | 5.84 (1.97) | 4.93 (2.42) | $t = 3.77$ ($p < 0.05$) |
| Financial Gains | 7.95 (2.07) | 8.37 (1.88) | $t = -2.17$ ($p < 0.05$) | 8.15 (1.91) | 7.80 (1.98) | $t = 1.46$ ($p = 0.15$) |
| Sustainability | 6.72 (2.33) | 6.88 (2.57) | $t = -0.55$ ($p = 0.58$) | 7.02 (2.12) | 5.93 (2.83) | $t = 3.04$ ($p < 0.05$) |

Mean values and standard deviation in brackets, and significant results are highlighted.

The perceptual difference between the two groups was found to be statistically significant after the treatments ($m_{2a} = 7.79$ (1.84), $m_{2d} = 6.10$ (2.35), $t = -4.01$, $p < 0.05$), which indicates that trust perceptions can be easily shaped if specific information is provided. When it comes to the intragroup comparison, both the advantages group ($m_{1a} = 6.81$ (2.24), $m_{2a} = 7.79$ (1.84), $t = -4.62$, $p < -0.05$) and the disadvantages group ($m_{1d} = 6.77$ (2.14), $m_{2d} = 6.10$ (2.35), $t = 2.55$, $p < 0.05$), showed significantly different attitudes following the manipulation and in the expected directions. This indicates that trust is a construct that can be relatively easily shaped.

*4.2. Security*

Cryptocurrencies introduce several new attack vectors and security concerns for companies and consumers alike [31]. Our items therefore included a general security perception as well as specific concerns pertaining to the buying, holding and transferring of cryptocurrencies. From a consumer perspective, financial security is paramount and was measured with a separate item.

Similar to trust, the difference between the groups in cryptocurrency security perceptions was found to be statistically significant after participants had watched the respective videos ($m_{2a} = 7.74$ (1.83), $m_{2d} = 6.26$ (2.65), $t = 3.25$, $p < 0.05$). When it comes to the intragroup comparisons, only the group that was exposed to the advantages of cryptocurrencies showed a significant attitudinal change ($m_{1a} = 7.03$ (2.31), $m_{2a} = 7.74$ (1.83), $t = -3.13$, $p < 0.05$), whereas the group that learned about potential security threats did not significantly change its attitudes ($m_{1d} = 6.78$ (2.33), $m_{2d} = 6.26$ (2.65), $t = 1.73$, $p = 0.09$).

*4.3. Privacy*

Previous research found that privacy concerns connected to the use of innovative technology can directly impact technology trust and indirectly decrease word-of-mouth and use as well use intentions via the constructs flow and perceived usefulness [32]. In our study, individuals' privacy attitudes were measured with two rather generic items that covered the general use of cryptocurrencies as well as the need to provide personal information to cryptocurrency exchanges.

The two videos produced the assumed effects; however, the difference between the two groups was not statistically significant following exposure to the respective informational videos ($m_{2a} = 6.06$ (2.65), $m_{2d} = 5.47$ (2.87), $t = 1.07$, $p = 0.29$), which indicates that privacy perceptions are harder to change than either trust or security perceptions. Similarly, the intragroup comparisons yielded a negligible attitudinal improvement for the group that watched the video on the advantages of cryptocurrencies ($m_{1a} = 6.05$ (2.59), $m_{2a} = 6.06$ (2.65), $t = -0.02$, $p = 0.98$) and a slight deterioration for the group that was exposed to the disadvantages ($m_{1d} = 6.18$ (2.35), $m_{2d} = 5.47$ (2.87), $t = -1.62$, $p = 0.11$); however, neither of these effects were statistically significant.

*4.4. Perceived Risk*

An early empirical study on the application of blockchain in marketing confirmed that risk aversion weakens the relationship between the usefulness of an online platform and

trust [33], thus, corroborating the relevance of consumers' risk perceptions. In our study, we used a four-item scale to cover various types of risks that individuals can be exposed to when using cryptocurrencies. Apart from a general risk assessment, we specifically asked participants whether they were concerned about theft or the risk that their transaction histories might be exposed. Furthermore, the intangible nature of cryptocurrencies might also exacerbate individual risk perceptions and was therefore included as an item. In order to be comparable with the other constructs, the scale for perceived risk was reverse coded so that higher values indicate lower (i.e., more positive) risk perceptions.

The two videos produced the assumed effects and the difference between the two groups was statistically significant following exposure to the respective stimuli ($m_{2a}$ = 6.87 (2.04), $m_{2d}$ = 4.93 (2.42), $t$ = 4.34, $p < 0.05$). The same is true for the pre and post comparisons, which revealed statistically significant results for the advantages ($m_{1a}$ = 6.21 (2.38), $m_{2a}$ = 6.78 (2.04), $t$ = −2.58, $p < 0.05$) as well as the disadvantages ($m_{1d}$ = 5.84 (1.97), $m_{2d}$ = 4.93 (2.42), $t$ = 3.77, $p < 0.05$) group, indicating that risk perceptions can be shaped relatively easily.

*4.5. Financial Gains*

Financial benefits are major drivers for cryptocurrency adoption, which can be attributed to their high volatility but also their potential to reduce transaction costs by removing intermediaries [2]. We used two items to measure perceived financial gains, including the expectation of high future yields and a general increase in value, which might affect, for example, the attractiveness of loyalty programs.

We did not detect a significant difference between the two groups in the post measurement ($m_{2a}$ = 8.37 (1.88), $m_{2d}$ = 7.80 (1.98), $t$ = 1.47, $p$ = 0.14). In the intragroup comparisons, however, the group that learned about potential advantages displayed a significantly positive attitudinal change ($m_{1a}$ = 7.95 (2.07), $m_{2a}$ = 8.37 (1.88), $t$ = −2.17, $p < 0.05$), while the attitude shift was non-significant in the group that was informed about potential financial losses ($m_{1d}$ = 8.15 (1.91), $m_{2d}$ = 7.80 (1.98), $t$ = 1.46, $p$ = 0.15). This implies that positive information about financial gains might be more influential to consumers than negative information regarding losses.

*4.6. Sustainability*

The final construct in our study is sustainability, which is of major importance for increasingly environmentally conscious consumers. Prior research established that, in lieu of specific information, consumers tend to infer product sustainability based on other product attributes [34]. The sustainability of cryptocurrencies is a fiercely discussed topic in the general media. The two items that we used to measure this construct included an assessment regarding their general sustainability as well as the potential impact of mining (i.e., the creation of new cryptocurrencies).

The attitudes of both groups changed after watching the respective videos; however, the difference between the groups was not statistically significant ($m_{2a}$ = 6.88 (2.57), $m_{2d}$ = 5.93 (2.83), $t$ = 1.75, $p$ = 0.08). The intragroup comparisons revealed a reversal of the pattern we found regarding attitudes towards financial gains. Informing study participants about cryptocurrencies' sustainability benefits did not result in statistically significant attitudinal changes ($m_{1a}$ = 6.72 (2.33), $m_{2a}$ = 6.88 (2.57), $t$ = −0.55, $p$ = 0.58); however, communication on the negative environmental impacts did produce a significant effect ($m_{1d}$ = 7.02 (2.12), $m_{2d}$ = 5.93 (2.83), $t$ = 3.04, $p < 0.05$).

**5. Discussion and Conclusions**

In this research, we investigated whether the provision of factually correct yet lopsided information can shape consumer attitudes toward cryptocurrencies. The findings from our experimental study indicate that attitudes can indeed be molded in this way and that some attitudinal constructs, all of which are important behavioral antecedents, might be more

easily affected than others. The results have substantial implications for academics and practitioners alike.

### 5.1. Theoretical Implications

The malleability of consumer attitudes toward novel technologies, such as cryptocurrencies, and the implications for marketing remains an under-researched topic in academia, as previous research has generated conflicting results on whether or not deep-rooted belief systems can be easily changed [27]. Previous research has already established the relevance of the topic by illustrating that individuals' information-processing biases can affect firm behavior, which in turn causes return anomalies [35]. In the context of marketing, researchers need to be mostly concerned about those behavioral antecedents that ultimately determine whether or not prospects will become buyers [23].

Novel technologies tend to substantially transform marketing activities, and numerous previous studies have already established the relevance of constructs, such as trust, security, privacy, perceived risk, financial gains and sustainability [1,2,18]. The question remains open regarding how malleable consumer attitudes towards these constructs actually are, yet this is especially important in the case of complex and constantly evolving technological transformation processes. Our findings provide the basis for future rigorous academic studies to investigate in more detail how attitudes toward new technologies are shaped and how attitudinal malleability can be incorporated into existing theories.

Furthermore, the gap between consumers' perceived knowledge and actual technological characteristics, as well as its relevance for buying decisions, presents a fruitful area for future research.

### 5.2. Practical Implications

For decades, marketers have been scrutinizing the consumer motivations underlying buying decisions. They use targeted advertising and companies' brands to actively shape the attitudes of their target groups. Novel technologies, such as cryptocurrencies, have already started to transform companies' marketing activities and brought about both new expectations and skepticism on the side of the consumers. It is therefore crucial for marketers to thoroughly understand a technology's potential and its perception among consumers.

More specifically, they not only need to identify the most important antecedents of consumer buying decisions but also comprehend whether the relevant attitudes are deep-rooted or rather easily manipulated. In this study, we provide initial evidence that some perceptions toward cryptocurrencies might be easier to change than others. In this regard, it also makes a difference whether the intended change is positive or negative. From an ethical perspective, this implies that it might be advisable for marketers to openly inform their target groups about their intended use of cryptocurrencies and the implications, which in turn might also create an atmosphere of trust. Finally, marketers need to closely attend to public discussions surrounding cryptocurrencies, which can affect consumer behaviors.

### 5.3. Limitations and Future Research

This experimental pilot study has a strong exploratory character. It was our goal to reveal whether the malleability of consumer attitudes toward novel technologies might constitute a relevant topic for marketers. Our findings imply that this is indeed the case. However, our study also has several limitations. To begin with, our sample size was relatively small, which implies a small statistical power. Contrariwise, this also implies that the true statistical significance might be even greater than we report, and our study may systematically underestimate the true importance of the topic.

The same is true for our choice of a convenience sample, which is an adequate means to uncover correlational changes but might not represent the general public. In the context of our study, however, this does not greatly matter since confounding effects on the relationships are eliminated through the experimental design. Again, the composition of the

sample with a strong tendency toward a younger, better-educated audience implies that the true effects might be underestimated since knowledge about cryptocurrencies might be less prevalent in the general population. Finally, the provision of the stimuli can be more balanced in future studies, and additional constructs need to be investigated.

In summary, our results lay a foundation for highly relevant future research. Most importantly, the findings need to be replicated across different consumer groups and could explore additional behavioral and attitudinal constructs. Furthermore, the investigation of other technologies is advisable, as is the assessment of whether individuals' subjective knowledge matches the objective features of the respective technology. Finally, in light of the proliferation of fake news, research is needed to determine the extent to which incorrect information can actually be used to shape consumer attitudes and how strongly attitudinal changes are connected to the possession of factual knowledge.

**Author Contributions:** Conceptualization, H.T. and E.G.; methodology, H.T. and E.G.; model validation, H.T.; review, E.G. and H.T.; data collection, E.G.; writing—original draft preparation, H.T.; writing—review and editing, H.T. All authors have read and agreed to the published version of the manuscript.

**Funding:** This research received no external funding.

**Informed Consent Statement:** Informed consent was obtained from all subjects involved in the study.

**Data Availability Statement:** Not applicable.

**Conflicts of Interest:** The authors declare no conflict of interest.

## Appendix A

| Construct | Variable | Modified Question |
|---|---|---|
| **Trust** | VAR_1.1 | Cryptocurrencies are trustworthy. |
| | VAR_1.2 | Even if cryptocurrencies are not fully regulated, I still trust them. |
| | VAR_1.3 | Generally, I trust cryptocurrency exchange systems. |
| **Security** | VAR_2.1 * | I am worried about owning cryptocurrencies because of their security. |
| | VAR_2.2 | I feel secure about buying, holding, and transferring cryptocurrencies. |
| | VAR_2.3 | Cryptocurrencies are secure for conducting financial transactions. |
| **Privacy** | VAR_3.1 * | When using cryptocurrencies my privacy is at risk. |
| | VAR_3.2 | I feel safe providing personal information to cryptocurrency exchange systems. |
| **Perceived * Risk** | VAR_4.1 | I feel at risk since I cannot touch or feel cryptocurrencies. |
| | VAR_4.2 | I am concerned about the potential of my cryptocurrencies being stolen. |
| | VAR_4.3 | The use of cryptocurrencies exposes me to a general risk. |
| | VAR_4.4 | If I use cryptocurrencies, hackers may be able to read my transaction history. |
| **Financial Gains** | VAR_5.1 | I believe cryptocurrencies will increase in value in the future. |
| | VAR_5.2 | Investing into cryptocurrencies will yield a high return on my investment. |
| **Sustainability** | VAR_6.1 | Cryptocurrencies have the potential to positively contribute to an environmentally friendly and sustainable society. |
| | VAR_6.2 * | Cryptocurrency mining has a negative impact on humanity. |

**Scale: 1 . . . totally disagree; 10 . . . totally agree. * reverse coded**

The items are based on [36–41] and were modified to fit the purpose of this study.

### Appendix B

| Topic | Link |
| --- | --- |
| **Advantages** | Benefits of cryptocurrencies. Available online: https://www.youtube.com/watch?v=nTo4iQYQuPs (Accessed on 1 June 2022). |
| **Disadvantages** | Problems of cryptocurrencies. Available online: https://www.youtube.com/watch?v=5-UO1t5EU90 (Accessed on 1 June 2022). |

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
