# Peer review of "On the Malleability of Consumer Attitudes toward Disruptive Technologies: A Pilot Study of Cryptocurrencies"

_information, doi:10.3390/info13060295_

Round 1

Reviewer 1 Report

Dear Authors, 

Thank you for the opportunity to read and review your manuscript submitted to Information. After reading the manuscript, I can clearly see that you have accomplished research that undoubtedly gives relevant findings. However, some issues stop me from being convinced that the quality of the current version is suitable for publication. Therefore, I advise major revision. Here is the list of the recommendations and comments:

  1. The Introduction should present the topic of the research more thoroughly. Please carefully review the current state of the research field, and mention the main aim of the research. 
  2. The Theoretical Background is too scarce. Are there any relevant theories related to malleability and consumer attitudes? 
  3. Please specify where the research was conducted. 
  4. Please explain why you have chosen to measure trust, security, privacy, perceived risk, financial gain, and sustainability? You present some rationale for these aspects in the research section, but the choice for the measurements should have a more substantial theoretical background explained in the literature review. 

Once again, thank you for the opportunity, and I wish you good luck in strengthening the manuscript.

Author Response

Reviewer 1

Dear Authors,  Thank you for the opportunity to read and review your manuscript submitted to Information. After reading the manuscript, I can clearly see that you have accomplished research that undoubtedly gives relevant findings. However, some issues stop me from being convinced that the quality of the current version is suitable for publication. Therefore, I advise major revision. Here is the list of the recommendations and comments.

Dear Reviewer,

First of all, we want to thank you for reviewing our paper and giving us important recommendations. In the table below we outline how we addressed them.

The Introduction should present the topic of the research more thoroughly. Please carefully review the current state of the research field, and mention the main aim of the research.  Thank you for letting us know that our introduction was insufficient. We rearranged the section such that the main goal of our research should be clear now. Furthermore, we end the introduction section with the research question that we investigate.
The Theoretical Background is too scarce. Are there any relevant theories related to malleability and consumer attitudes? 

We added some additional information on the theoretical background of (attitudinal) malleability. In a nutshell, there are two theoretical “families”, namely entity theory and incremental theory. We now mention this in the paper and added some further empirical studies.

Please specify where the research was conducted. 

The research was conducted online using a snowballing approach for sampling. We now mention this in the paper.

Please explain why you have chosen to measure trust, security, privacy, perceived risk, financial gain, and sustainability? You present some rationale for these aspects in the research section, but the choice for the measurements should have a more substantial theoretical background explained in the literature review. 

In a nutshell, those were the major attributes that we identified in the literature as drivers of individuals’ decision to adopt cryptocurrencies. We added a paragraph in the “Theoretical Background” section where we elaborate on this in more detail and also included additional references.

Once again, thank you for the opportunity, and I wish you good luck in strengthening the manuscript

Thank you so much for your constructive feedback! We hope that we were able to address all your concerns sufficiently.

Reviewer 2 Report

Regarding the content, I do not have any changes to recommend, it makes a good literary review to support the relevance of the problem to be studied and a good structuring of the content, it uses the correct methodology for this type of study and it is a consistent and well-detailed methodology to give significance to the results they show, makes a good discussion of the results with respect to the studies carried out previously, and marks the conclusion obtained well.

Although I advise looking at these things:

Eliminate anonymity in the article: line 7 and sections “Supplementary Materials” and “Author Contributions”.

On line 107, you should put “Appendix A”, not “Appendix”.

And in the section “5. Discussion and conclusion”, you must also include a comparison of the results achieved with the results achieved in previous studies, citing and referencing said studies.

Author Response

Reviewer 2

Regarding the content, I do not have any changes to recommend, it makes a good literary review to support the relevance of the problem to be studied and a good structuring of the content, it uses the correct methodology for this type of study and it is a consistent and well-detailed methodology to give significance to the results they show, makes a good discussion of the results with respect to the studies carried out previously, and marks the conclusion obtained well.

Dear Reviewer,

First of all, we want to thank you for reviewing our paper and giving us important recommendations. We are happy to hear that you are satisfied with the content of our paper.

Although I advise looking at these things:

Eliminate anonymity in the article: line 7 and sections “Supplementary Materials” and “Author Contributions”.

We removed the anonymity in these sections.
On line 107, you should put “Appendix A”, not “Appendix”.

Following your suggestion we relabeled it into Appendix A. Furthermore, we added another appendix in which we included the video links.

And in the section “5. Discussion and conclusion”, you must also include a comparison of the results achieved with the results achieved in previous studies, citing and referencing said studies.

Thank you for letting us know that this important part was missing. We now compare our study with previous research.

Finally, let us thank you again for taking the time to review our paper.

--- The authors

Reviewer 3 Report

The article addresses an interesting and timely topic related to the use of cryptocurrencies in marketing activities. The purpose of the study is to determine whether people's attitudes toward cryptocurrencies can change in the short term if they are provided with true but biased information. The authors designed a scientific experiment by dividing study participants into two groups and showing them a short video with true but unbalanced information about the advantages and disadvantages of cryptocurrencies. One group was presented the advantages alone, while the other group was presented the disadvantages alone. People's attitudes about trust, security, privacy, perceived risk, financial gains, and sustainability were measured both before and after watching the video. It was found that positive information improved individuals' attitudes, while negative information had the opposite effect. Thus, it was found that consumer attitudes can be quite malleable, but to varying degrees depending on the construct chosen; some more easily than others. The authors are aware of the limitations of the study conducted due to the lack of representativeness of the sample and point this out in the Discussion and Conclusions section.

In order for the article to be published in Information journal, the following issues should be clarified:

  1. The concept of true but biased information about cryptocurrencies needs to be better clarified. It is not enough to just highlight their advantages and disadvantages. Basically, the reader has no idea about the information that affects the attitudes of the study participants. Therefore, I believe that the two videos should be published along with the article. Their transcription can also be included in the article as an appendix. In essence, the authors are investigating the psychological phenomenon of inducing waves of pessimism and optimism in the minds of experimental participants, which has already been well explained by John M. Keynes in his The General Theory of Employment, Interest and Money (1936). In other words, the authors do not state exactly what stimuli affect the respondents.
  2. The limited usefulness of the study results is due to the small and unrepresentative sample. It consists of 100 individuals, predominantly young and well-educated males. Moreover, these individuals are well acquainted with the issue under study, as two-thirds of them own cryptocurrencies. The relative ease with which their attitudes can be manipulated suggests that for a broader group of consumers, these attitudes could be even more flexible. This certainly adds to the uncertainty and complexity of cryptocurrency markets.
  3. The lack of representativeness of the sample indicates that we are dealing with a pilot study. This fact must be signaled already in the title and the executive summary.
  4. In my opinion, the study, besides being limited, is also incomplete. The authors did not use all the possibilities at their disposal. Yes, different videos were shown to two groups of participants and some interesting, albeit partial, results were obtained. Unfortunately, they were largely predictable, as it is obvious that good information evokes optimism and bad information evokes pessimism. However, the study could be expanded so that all participants could be shown both movies and attitudes measured before and after. This could be more objective. It would also be interesting to compare the two approaches. The authors should address these issues at least in theory if repeating the experiment would prove impossible.

Author Response

Reviewer 3

The article addresses an interesting and timely topic related to the use of cryptocurrencies in marketing activities. The purpose of the study is to determine whether people's attitudes toward cryptocurrencies can change in the short term if they are provided with true but biased information. The authors designed a scientific experiment by dividing study participants into two groups and showing them a short video with true but unbalanced information about the advantages and disadvantages of cryptocurrencies. One group was presented the advantages alone, while the other group was presented the disadvantages alone. People's attitudes about trust, security, privacy, perceived risk, financial gains, and sustainability were measured both before and after watching the video. It was found that positive information improved individuals' attitudes, while negative information had the opposite effect. Thus, it was found that consumer attitudes can be quite malleable, but to varying degrees depending on the construct chosen; some more easily than others. The authors are aware of the limitations of the study conducted due to the lack of representativeness of the sample and point this out in the Discussion and Conclusions section.

Dear Reviewer,

First of all, we want to thank you for reviewing our paper and giving us important recommendations. In the table below we outline how we addressed them.

Your summary nicely wraps up the content and the intention of our study.

In order for the article to be published in Information journal, the following issues should be clarified:The concept of true but biased information about cryptocurrencies needs to be better clarified. It is not enough to just highlight their advantages and disadvantages. Basically, the reader has no idea about the information that affects the attitudes of the study participants. Therefore, I believe that the two videos should be published along with the article.Their transcription can also be included in the article as an appendix. In essence, the authors are investigating the psychological phenomenon of inducing waves of pessimism and optimism in the minds of experimental participants, which has already been well explained by John M. Keynes in his The General Theory of Employment, Interest and Money (1936). In other words, the authors do not state exactly what stimuli affect the respondents. Thank you for letting us know that this important information was missing. We included the links to the two videos in Appendix B. In the videos we outline various aspects of cryptocurrencies and also use survey data to illustrate our respective points. We hope that the inclusion of the videos makes our research approach more transparent.
The limited usefulness of the study results is due to the small and unrepresentative sample. It consists of 100 individuals, predominantly young and well-educated males. Moreover, these individuals are well acquainted with the issue under study, as two-thirds of them own cryptocurrencies. The relative ease with which their attitudes can be manipulated suggests that for a broader group of consumers, these attitudes could be even more flexible. This certainly adds to the uncertainty and complexity of cryptocurrency markets.

We fully agree with you that our results indicate that the attitudes of a less knowledgeable user groups might be even more malleable. We explicitly mention this now in our limitation section and hope that future studies will investigate attitudinal malleability in a wider population. Furthermore, we now make it clear that this is an explorative study that is based on a convenience sample.

The lack of representativeness of the sample indicates that we are dealing with a pilot study. This fact must be signaled already in the title and the executive summary.

We now mention that this is an explorative study in the title. Additionally, we added the information that we used a convenience sample in the abstract.

In my opinion, the study, besides being limited, is also incomplete. The authors did not use all the possibilities at their disposal. Yes, different videos were shown to two groups of participants and some interesting, albeit partial, results were obtained. Unfortunately, they were largely predictable, as it is obvious that good information evokes optimism and bad information evokes pessimism.

The goal of our study was to investigate how malleable the attitudes of the participants actually were. Among other factors, this might depend on the previous level of knowledge and their self-consciousness. While it might seem predictable that positive information evokes optimism (and vice versa) our results illustrate that this effect is only significant for some of the constructs that we investigated.

However, the study could be expanded so that all participants could be shown both movies and attitudes measured before and after. This could be more objective. It would also be interesting to compare the two approaches. The authors should address these issues at least in theory if repeating the experiment would prove impossible.

While such a study design would be indeed possible, it would be more feasible in a longitudinal study with some time between the different treatments. Since this was a cross-sectional study, we decided to focus on one treatment respectively and accounted for potential biases by having a relatively large sample size for an experimental study and by using randomized treatment assignment.

Finally, let us thank you again for reviewing our paper and providing us with valuable feedback.

Kind regards,
--- The authors

Reviewer 4 Report

This is an exciting and relevant paper comparing people's approach toward cryptocurrencies after experiencing data manipulations. I suggest comparing these results to prior work on biased information in the financial field in general. see for example Biased beliefs, asset prices, and investment:structural approach, Alty and Tetlock Journal of finance,69,1. I also suggest using more figures to describe the results for more contingency for readers. 

Author Response

Reviewer 4

This is an exciting and relevant paper comparing people's approach toward cryptocurrencies after experiencing data manipulations. I suggest comparing these results to prior work on biased information in the financial field in general. see for example Biased beliefs, asset prices, and investment:structural approach, Alty and Tetlock Journal of finance,69,1. I also suggest using more figures to describe the results for more contingency for readers.  Dear Reviewer,First of all, we want to thank you for reviewing our paper and giving us important feedback. We are happy that you find our paper to be exciting and relevant. Thank you for letting us know that we missed a seminal publication. We integrated it into our discussion section. As far as the figures are concerned, we refrained from including additional visuals, since our paper already contains a very comprehensive figure and three tables. We hope that you find our decision acceptable.Thank you very much!--- The authors

Round 2

Reviewer 1 Report

Dear Authors, 

Thank you for the revisions. The current version of the manuscript looks much better, however, there is one minor comment that needs to be addressed. It is praiseworthy that you have made changes in the Introduction, however, it is still insufficient. Please carefully review the current state of the research field, and mention the main aim of the research.

Author Response

Reviewer 1

Dear Authors,  Thank you for the revisions. The current version of the manuscript looks much better, however, there is one minor comment that needs to be addressed. It is praiseworthy that you have made changes in the Introduction, however, it is still insufficient. Please carefully review the current state of the research field, and mention the main aim of the research.

Dear Reviewer,

First of all, we want to thank you for reviewing our paper again and giving us important recommendations. We are happy to hear that you find that our manuscript now looks much better. In order to address your concern, we added several more references in the introductory section and tried to make it clearer what our actual goal is. We hope that that you find the revised version to be publishable.

Kind regards,

--- The authors

Reviewer 3 Report

The authors have included links to the videos that support the research conducted. In addition, they have responded to comments made in a previous review, but these responses are not entirely satisfactory. The following is my opinion.

1. After watching both videos, my observations are as follows:

A. As I suspected (while writing the first review), the authors made some mistake in defining the information provided to the study participants. They claim that the information is true, but biased. This is not the case. In the first video, optimistic forecasts of cryptocurrency market growth through 2030 are presented. As we know, the forecasts are neither true nor false. It is an induction of waves of optimism, and the result is very easy to predict.

B. The source of these forecasts is not given, and as we know this is important in financial markets. Some sources may be more reliable and others less so.

C. The information provided in the two videos is not symmetrical. The first video contains optimistic forecasts for the development of the cryptocurrency market until 2030. In order to be balanced, the second video should also contain forecasts for the development of cryptocurrency risks in a similar time frame.

D. Furthermore, even if true information is provided, it is not complete. For example, in a video on cryptocurrency problems, it is reported that almost all cryptocurrency exchanges were hacked in 2020, but it is not stated how many of these hacks were successful. That is, it is not clear what the security level of transactions is.

2. I made it clear in a previous review that the sample is small and unrepresentative. The implication is that this is a pilot study and that this fact should have been signaled in the paper. Instead, the authors write that they use EDA (exploratory data analysis). This is not the same thing. The characteristic feature of EDA is that it does not formulate and test research hypotheses. This condition is met here. Nevertheless, the study conducted by the authors has all the characteristics of a pilot study and this should be clearly indicated to the reader.

3. All of the variables analyzed by the authors are qualitative, so a correspondence analysis would work much better here.

In such a situation, the question arises what to do next with the article. Some minor mistakes were made at the research stage, which basically cannot be undone anymore. Considering the overall good evaluation of this work, which I emphasized in the previous review, I propose to publish the article after clarifying in the text the issues raised in the above points.

Author Response

Reviewer 3

Dear Authors,  The authors have included links to the videos that support the research conducted. In addition, they have responded to comments made in a previous review, but these responses are not entirely satisfactory. The following is my opinion. 1. After watching both videos, my observations are as follows: A. As I suspected (while writing the first review), the authors made some mistake in defining the information provided to the study participants. They claim that the information is true, but biased. This is not the case. In the first video, optimistic forecasts of cryptocurrency market growth through 2030 are presented. As we know, the forecasts are neither true nor false. It is an induction of waves of optimism, and the result is very easy to predict.

Dear Reviewer,

First of all, we want to thank you for reviewing our paper again and giving us important recommendations.

We fully understand your point regarding the forecasts. Our wording was indeed wrong and we changed that. We added the information that we also included market forecasts. We agree with you that these are neither true nor false, but we wanted to make it clear that we did not fabricate any information for the purpose of our study. We hope that we were able to make this clear in our revised version and we also added the source.

B. The source of these forecasts is not given, and as we know this is important in financial markets. Some sources may be more reliable and others less so.

Our information was taken by a report from a company called Facts and Factors. You are absoutely right in that some market researchers might be more reliable than others. However, for the purpose of our study exxaggerated forecasts might even be benefical (i.e., to better understand how malleable individuals’ attitudes are). However, we fully understand that we need to disclose the information that we used in the videos. Therefore we included the source in the paper and also added the fact that we used forecasts from market researchers.

C. The information provided in the two videos is not symmetrical. The first video contains optimistic forecasts for the development of the cryptocurrency market until 2030. In order to be balanced, the second video should also contain forecasts for the development of cryptocurrency risks in a similar time frame.

You raise a good point. Actually, we searched quite some time to find some “negative” information regarding cryptocurrency market developments, but could not find any. While there are several predictions that Bitcoin might collapse, we did not find a serious study that predicts a substantial downturn for the whole market in the medium term.

D. Furthermore, even if true information is provided, it is not complete. For example, in a video on cryptocurrency problems, it is reported that almost all cryptocurrency exchanges were hacked in 2020, but it is not stated how many of these hacks were successful. That is, it is not clear what the security level of transactions is.

Thank you again for letting us know that this was unclear. It was indeed our goal to see how biased information (i.e., information that is not put into perspective) might potentially impact individuals’ attitudes. After all, this is what frequently happens in the public media. It was just our intention to use “true” information (or market forecasts) for this purpose. We hope that we were able to make this clear in our revised version.

2. I made it clear in a previous review that the sample is small and unrepresentative. The implication is that this is a pilot study and that this fact should have been signaled in the paper. Instead, the authors write that they use EDA (exploratory data analysis). This is not the same thing. The characteristic feature of EDA is that it does not formulate and test research hypotheses. This condition is met here. Nevertheless, the study conducted by the authors has all the characteristics of a pilot study and this should be clearly indicated to the reader.

Following your recommendation, we now make it clear that we conduct a pilot study.

3. All of the variables analyzed by the authors are qualitative, so a correspondence analysis would work much better here.

The goal of a correspondence analysis is to display a data set in a two-dimensional form and to find a structure in categorical data. Since we relied on existing scales (albeit modified) we did not question the underlying data structure but rather analyzed them in a quantative manner.

In such a situation, the question arises what to do next with the article. Some minor mistakes were made at the research stage, which basically cannot be undone anymore. Considering the overall good evaluation of this work, which I emphasized in the previous review, I propose to publish the article after clarifying in the text the issues raised in the above points.

We are grateful for your overall positive assessment. We agree that there were several minor flaws that we overlooked in the research design and especially the provision of the stimuli. We included them in the limitations and hope that future researchers will build on our work and avoid our shortcomings. Again, thank you for your willingness to review our paper twice and provide us with helpful feedback.

--- The authors